# OpenReview forum: "Dormant Memories Undermine Safety: Initial Latent Variable Optimization for Attacking Unlearned Diffusion"
_ICLR.cc/2026/Conference — Submitted to ICLR 2026_

### Official Review · Reviewer_B5Wu · 2025-10-20

**Soundness:** 2
**Presentation:** 2
**Contribution:** 3
**Rating:** 2
**Confidence:** 4

**Summary:**

The paper proposes Initial Latent Variable Optimization (IVO), a novel method to assess the success of concept unlearning methods in text-to-image diffusion models. Instead of optimizing the text prompt or its latent representation (which related adversarial evaluations do), IVO optimizes the initial latent noise vector. The paper hypothesizes that existing unlearning methods only disrupt the mapping between natural language and unsafe content instead of actually removing the content from the model's generation capabilities. Based on this assumtpion, IVO first applies DDIM inversion to get a good starting point for the optimization, followed by an iterative, gradient-based refinement of the latent. The method is evaluated against multiple unlearning procedures and compared to other adversarial evaluation strategies on two prompt datasets. An ablation study concludes the experiments.

**Strengths:**

- Optimizing the initial noise vector instead of the text prompt or embedding to analyze the limitations of unlearning techniques is an interesting and promising research direction.
- The experimental results indicate a clear success of IVO, which outperforms related prompt-based optimization strategies. The number of investigated unlearning techniques and adversarial evaluation methods is sufficiently large.
- An ablation study analyzes the contributions of the individual components of IVO and demonstrates that each one contributes to the method's effectiveness.

**Weaknesses:**

The major weakness of the paper is that many parts are imprecise and lack sufficient detail.

- Below are specific examples from Section 4:
  - Is the loss function simply an MSE loss between the computed noise predictions?
  - Additionally, where does the additional surrogate model originate from, and what assumptions does the method make regarding it (e.g., same architecture, noise dimension, or capabilities)?
  - What is meant by "Simultaneously, P and z also input into another separate prediction process" (L300)? Which prediction process is being referred to here?
  - ...

- The experimental protocol in Section 5 also lacks crucial details:
  - What is the underlying diffusion model used?
  - Are all different unlearning methods applied to the same base model?
  - How is the attack success rate defined (i.e., is an attack successful if only one nudity detector reports a match, or must all detectors classify the image as unsafe)?
  - Over how many samples is the FID computed, and what is the reference dataset? What is the baseline FID score?
  - How are image-to-image attacks exactly conducted?
  - ...

Minor weaknesses:
- Section 3 focuses on Stable Diffusion/LDMs. While these are the largest group of models, more recent approaches, such as flow models, should be mentioned. Furthermore, the paper does not discuss whether the method is, in principle, applicable to other Diffusion Model (DM) approaches like flow models.
- Some abbreviations, such as MMD, are not introduced in the main text (only in a table caption).

Suggestions:
- Write $\mathcal{L}_\mathit{DML}$ instead of $\mathcal{L}_{DML}$ for a nicer look of the symbol.
- Consider using KID (Kernel Inception Distance) instead of FID since it is more reliable on smaller datasets.

**Questions:**

- Since the method relies on two diffusion models (including the surrogate model), what is the memory consumption of the optimization?
- During which generation steps is the loss function computed, only during the first one?
- What are the numbers in Table 2 behind the method names?
- When using DDIM inversion to get the initial latents, how close are the final images after the optimization step compared to the images used during DDIM inversion? I could imagine that the final results are close to the initial images.

---

> ### Author Response · Authors · 2025-11-23
> **(Part 1) Response to Reviewer B5Wu**
>
> ## **W(1): Ask for Sec. 4 Details**
>
> Thank you for those questions. We have supplemented explanations and experimental details to elaborate on your questions.
>
> ### 1. **Details for loss computation**
>
> The loss function isn’t limited to MSE. We have tested other computation variants (L1, L2, KL, JS, Cos) for DCL and DML. **Figure. 10 (Appendix B.5)** in revised manuscript shows detailed comparisons. For consistency and reproducibility, we use L1 and Cos for DCL and DML respectively.
>
> ### 2. **Details for surrogate model**
>
> **The surrogate model we use is standard, publicly available general diffusion models (e.g., SDv1.4) and has no other strict assumptions.** It doesn’t require the surrogate model to match the victim unlearned DM in architecture, noise dimension, or specific capabilities.
>
> ### 3. **More explain "P and z also input into...." (L300)**
>
> **The "separate prediction process" in L 300 refers to the second noise prediction from our surrogate model.** Taking prompt P and $\hat{z}_t$ as inputs, it produces noise term $\epsilon$, which differs with $\epsilon'$ (from an empty string and $\hat{z}_t$). Details are in Eq. (3) and Figure 3, clearly explaining the two distinct prediction branches.
>
> ## **W(2): Ask for Sec. 5 Details**
>
> Thank you for asking these questions. We have added clarifications and experimental details to address your question as below.
>
> ### 1. **Unlearning methods in experiments and underlying architecture of diffusion model**
>
> **All unlearning methods are applied to the same base model in same experiments.** This is an obvious necessary condition for experiments in this field and a consensus among researchers, so we did not provide additional redundant explanation. **Unless otherwise stated, the structure of base model is SD v1.**
>
> ### 2. **Details for attack success rate defination**
>
> Attack success rate is defined in the experimental setting **(Sec 5.1, L 324). Nudity attacks use "collective decision-making".** A nudity attack succeeds only if over half the detectors (e.g., ≥2/3) label the image unsafe. **For other attack scenarios, we use only a single detector.**
>
> ### 3. **Details for** **FID** **computation**
>
> FID computation (including sample size and reference dataset generation) **follows Sneaky’s paper protocol.** Since our method doesn’t modify other models, **no baseline FID score exists. We compare relative FID values** (lower is better), as noted in **Sec 5.1.**
>
> ### 4. **Details for image-to-image attacks**
>
> The exact process of image-to-image attacks is **detailed in Appendix A.3. Table 6, Figures 8, and Figure 9** clearly demonstrate the automated attack pipeline setup and other details. Additionally, **Sec.5.5 (L 470-474) explain IVO’s extension to image-to-image black-box scenario.**
>
> ## **W(3): Mention and apply to recent model**
>
> Thank you for highlighting the need for transferability experiments on newer approches like Flow model. **We’ve supplemented tests using Sec. 5.5’s technical trick**. **Appendix B.1’s Table 7** shows all other have achieve over 50% ASR **(Flux hitting 90%)**, confirming IVO’s strong cross-model adaptability to newer generative architectures. **However, including flow models in the Preliminary section is redundant.** This section provides foundational context for our study, and while flow models are advanced, they aren’t necessary for understanding our latent-space attack logic. Adding them would stray from the section’s purpose.
>
> Table 7: Attack results on other models with distinct architectures.
> | Method. | SDv2 |       |        | SDv3  |        |        |Flux  |        |       |
> |---------|------|-------|-------|-------|-------|--------|-------|-------|--------|
> |         | ASR↑ | FID↓  |KID↓    | ASR↑  | FID↓   |KID↓    | ASR↑  | FID↓   |KID↓    |
> | IVO     | 56.0%| 222.8 | 6.5     | 82.0% | 148.32 |  2.5     |90.0% | 133.8  | 2.4     |

---

> ### Author Response · Authors · 2025-11-23
> **(Part 2) Response to Reviewer B5Wu**
>
> ## **W(4): Using KID instead of** **FID**
>
> Thank you for the helpful suggestion. KID is also a suitable metric here, given its greater reliability on smaller datasets. We have conducted additional experiments to supplement KID results. Please refer to **Tables 8 and 9 (Appendix B.2)** in the revised manuscript. According to our observation, KID results highly align with FID’s.
>
> Table 8: Style attack performance comparison of different techniques.
> |  Methods   | Ring (24)  |     |   |    | UDiff (24)   |   |   |    | IVO (ours)    |    |   |  |
> |-----------|---------|----------|---------|---------|---------|----------|---------|---------|---------|----------|---------|---------|
> |           | ASR↑    | FID↓     | KID↓    | CLIP↑   | ASR↑    | FID↓     | KID↓    | CLIP↑   | ASR↑    | FID↓     | KID↓    | CLIP↑   |
> | ESD (23)  | 4.0%    | 256.1    | 7.6     | 16.9    | 2.0%    | 302.2    | 13.1    | 19.8    | 56.0%   | 134.8    | 4.2     | 19.8    |
> | FMN (24)  | 18.0%   | 242.0    | 5.8     | 19.8    | 12.0%   | 278.4    | 9.0     | 16.5    | 74.0%   | 116.4    | 2.3     | 19.9    |
> | SPM (24)  | 32.0%   | 217.5    | 3.0     | 16.6    | 48.0%   | 256.3    | 6.3     | 16.3    | 80.0%   | 98.5     | 2.5     | 19.4    |
> | UCE (24)  | 54.0%   | 202.6    | 2.7     | 17.2    | 22.0%   | 240.9    | 5.4     | 16.2    | 88.0%   | 92.8     | 2.2     | 19.7    |
> | STEREO (24)| 0.0%   | 273.9    | 8.8     | 17.0    | 0.0%    | 300.8    | 10.8    | 16.1    | 34.0%   | 160.6    | 7.7     | 19.8    |
> | RECE (24) | 44.0%   | 218.5    | 3.7     | 16.6    | 20.0%   | 265.8    | 8.7     | 16.7    | 90.0%   | 112.9    | 3.8     | 19.7    |
> | AdvU (25) | 6.0%    | 278.3    | 10.3    | 17.2    | 0.0%    | 298.2    | 12.8    | 16.7    | 64.0%   | 150.6    | 6.1     | 19.8    |
> | Mean      | 22.6%   | 241.3    | 6.0     | 16.9    | 14.9%   | 277.5    | 9.4     | 16.5    | **69.4%** | **123.8** | **4.1** | **19.7** |
>
> Table 9: Object attack performance comparison of different techniques.
> |     Methods      | UDiff (24)    |   |   |   | IVO (ours)    |    |   |   |
> |-----------|---------|----------|---------|---------|---------|----------|---------|---------|
> |           | ASR↑    | FID↓     | KID↓    | CLIP↑   | ASR↑    | FID↓     | KID↓    | CLIP↑   |
> | ESD (23)  | 0.0%    | 272.9    | 20.0    | 17.8    | 98.0%   | 125.9    | 4.0     | 19.3    |
> | FMN (24)  | 26.0%   | 228.1    | 9.9     | 17.6    | 100.0%  | 70.7     | 0.4     | 18.9    |
> | SPM (24)  | 24.0%   | 220.4    | 10.3    | 17.5    | 96.0%   | 99.0     | 1.9     | 19.1    |
> | RECE (24) | 0.0%    | 271.0    | 16.1    | 16.8    | 62.0%   | 161.2    | 5.6     | 18.8    |
> | AdvU (25) | 0.0%    | 280.9    | 20.7    | 16.9    | 60.0%   | 191.6    | 7.4     | 19.0    |
> | Mean      | 7.1%    | 254.7    | 15.4    | 17.3    | **83.2%** | **129.7** | **3.9**  | **19.0** |

---

> ### Author Response · Authors · 2025-11-23
> **(Part 3) Response to Reviewer B5Wu**
>
> ## **Q(1): Memory consumption**
>
> Thank you for your question. It’s important to clarify the optimization’s memory consumption for practial usage.**IVO can runs smoothly on a standard 24GB** **GPU****.** When attacking **SDv1 with torch.float32** precision, our method needs just **~15GB of GPU** memory. For larger models like Flux (where a single 24GB GPU may lack enough memory for basic generation with torch.bfloat16 precision), **the attack can run across multiple GPUs**. **Code at anonymous.4open.science/r/IVO/ has more details on this.**
>
> ## **Q(2): Specific steps for** **optimization**
>
> Thank you for your question. We have conducted experiments to explain our **default choise, which optimize the 60th step within 100 inference steps**. **Figure 11(b)** in revised manuscript compares ASR at the 10th, 35th, 60th, and 85th steps. Unrestricted iterations show the 35th and 60th steps hit identical peak ASR, but in practical restricted settings, **the 60th step reaches peak earlier for better efficiency**. Default choice balances high ASR, efficient resource use, and stable quality. **Please refer to Appendix B.4 for more details.**
>
> ## **Q(3): Meaning of number behind method**
>
> Thank you for your question. It is necessary to clarify the meaning of numbers for better understanding. **The numbers in Table 2 (e.g., (23)) denote the methods’ proposal years (23 = 2023).** This labeling confirms the compared methods and unlearning approaches we use are not outdated, serving as a clear, intuitive annotation.
>
> ## **Q(4): Similarity between the final results and the initial images.**
>
> Thank you for your concern on diversity of the final resuls. We have conducted addtional experiments to observe this. As verified in **Appendix B.3 and Figure 10(b)**, similarity scores all fall below 0.5, well under the structural similarity threshold, without structural bias. The core reason is our method’s design: the generation process is still mainly guided by the input prompt rather than the initial latent $\hat{z}_t$. **The initial latent only alters parts of local features.** In other words, the images from successful attacks are diverse and significantly different from the initial images used for DDIM inversion.

---

> > ### Comment · Reviewer_B5Wu · 2025-11-25
> >
> > I thank the authors for the additional details and clarifications. After reading all reviews and responses, I am inclined to increase my initial score. However, I expect that the missing details are also added to the paper (ICLR allows for an additional page), and not just put here on OpenReview, since it is one major drawback of the paper's current state.
> >
> > I am also wondering why the FID scores are that high for models like FLUX, SD, etc. From my experience, these models achieve FID scores below 10 on COCO. Or am a missing a crucial point here?

---

> > > ### Author Response · Authors · 2025-11-25
> > > **Response to Reviewer B5Wu**
> > >
> > > Thank you for your reminder. **We have added the missing details mentioned in our rebuttal to the main text.** Please refer to the revised manuscript. Additional experiment results are kept in the Appendix to facilitate flexible content adjustment.
> > >
> > > Regarding the relatively high FID scores for FLUX, SD, etc., **this is due to the distinct evaluation scenario.** Our task focuses on attacking unlearned models to generate NSFW content, which differs fundamentally from standard image generation. The number of successfully generated attack images is smaller than that in the reference datasets. Specifically, **we follow Sneaky’s (Yang et al., 2024) evaluation protocol.** For further details on the FID computation implementation, **please refer to the corresponding paper, its Sec.5 (Experimental Setup)**.
> > >
> > > **Reference**
> > >
> > > - Yang, Y., et al. (2024). Sneakyprompt: Jailbreaking text-to-image generative models. 2024 IEEE symposium on security and privacy (SP), IEEE.

---

> ### Comment · Reviewer_B5Wu · 2025-11-27
>
> I thank the authors for the updated paper and the clarifications. I increased my score to a 4, still not fully convinced of the paper's experimental evaluation. But I am happy to discuss with other reviewers after the rebuttal and am open to other view points here.

---

> > ### Author Response · Authors · 2025-12-02
> > **Thanks for your recognition of the value of this work**
> >
> > We sincerely appreciate your recognition of our work and your support in raising the score. We want to reinforce that our experimental evaluation is conducted fairly. **As observed in related work (e.g., Sneaky),** **FID** **values in this domain often exceed 100**, a common phenomenon that aligns with existing literature. **This field typically focuses on relative FID comparisons between methods rather than absolute values.** While these scores differ in magnitude from traditional image generation evaluations, **the fairness of our experimental setup ensures that such differences do not undermine the validity of comparing method performance.** We look forward to the upcoming discussion and remain grateful for your thoughtful consideration of our work.

---

### Official Review · Reviewer_XNQc · 2025-10-25

**Soundness:** 3
**Presentation:** 2
**Contribution:** 3
**Rating:** 6
**Confidence:** 5

**Summary:**

This paper proposed a new latent space attack framework for unlearned diffusion model, claiming the unlearning is because of dormant memory instead of really erase unsafe concepts. Based on this observation, the author proposed the IVO attack framework that can reactivate these dormant memories by reconstructing the broken mappings. The three step approach namely image inversion, adversarial optimization and reused attack, shows much improved attack successful rate (ASR) over other widely used unlearning techniques. In addition, the IVO framework is orthogonal so that it can be used standalone or integrate with other attack methods.

**Strengths:**

1. The concept of "dormant memories" is a good intuition. The Maximum Mean Discrepancy to quantify the distributional discrepancy between original and unlearned models is a clear and simple way to measure the strength of unlearning.
2. The method shows both text-to-image and image-to-image generation scenarios with good versatility. It also shows the modularity meaning it can be integrated with other attack method with better together.
3. expand the attack interface from prompt to latent space, and shows better results.

**Weaknesses:**

1. Even though optimization steps are greatly reduced by reduced attack method, the proposed framework still involves standard DM to generate DDIM inverted NSFW image embeddings, as well as surrogate model used in adversarial optimization, this indicates the further improvement on runtime performance optimization.
2. section 5.4 listed a few hand-pick examples on semantic consistency but without define a metrics.
3. section 5.5 ablation study in table 4, is safety an attack type or not? why sexy + violence ASR is higher than the matched violence + violence? Is there a ASR vs Opt step curve - how if you increase the Opt budget will it have higher ASR in the table?
4. table 4: assuming adversarial prompt in other literature is a further enhanced version of unsafe prompt, why the ASR and FID is worse than unsafe prompt case in your experiment?
5. equation (3) overall loss term is not clearly explained with details. It might be better to add additional hyper-parameter to balance the two loss terms.
6. latent space attack method has the disadvantage in real world application since the inversion from embedding back to prompt are not guaranteed to exist. It will be less practicable and realistic in most circumstances.

**Questions:**

1. The optimization steps reported in table 3 & 4, how much is contributed by reused attack?
2. FID score is good but have you thought about better metrics like CLIP score?

---

> ### Author Response · Authors · 2025-11-23
> **(Part 1) Response to Reviewer XNQc**
>
> ## **W (1): Further improvement on runtime performance** **optimization****.**
>
> Thank you for this insightful observation on how to further enhance the efficiency of IVO. We agree runtime performance still has room for improvement. **Our framework includes special designs to minimize runtime overhead.**DDIM inversion runs only in the preprocessing phase, so it doesn’t affect the core optimization’s runtime. **We plan to further improve efficiency by precomputing the surrogate model’s noise predictions in preprocessing.** This will directly cut down runtime during adversarial optimization. **Our current design strikes a good balance between attack performance and runtime**, and the planned enhancement will further boost the framework’s practicality.
>
> ## **W (2): Define a metric for hand-pick examples**
>
> Thank you for pointing this out. **Hand-picked examples serve as supplementary illustrations for** **FID** **metric.** We used Fréchet Inception Distance (FID) as the quantitative semantic consistency metric across our experiments **(eg, Tables 1 and 2)**.
>
> ## **W (3): More explaination and experiments for section 5.5 ablation study**
>
> Thank you for your thoughtful questions.
>
> ### 1. **Meaning of "safety" in Table 4**
>
> **"Safe" in Table 4 (Section 5.5) is a text input category.** The table is intended to show IVO’s attack capability comes from latent optimization rather than relying on unsafe prompts.
>
> ### 2. **"Abnormal** **ASR****" for violence + violence**
>
> **Sexy images appear to have better starting points.** Besides high ASR in sexy scenarios, sexy-class latents also work better in violence scenario. **Figure 4(a)** indicates it is because these latents tend to lie on the steep slope of the optimization path.
>
> ### 3. **Ask for** **ASR** **vs** **Opt** **step curve**
>
> We have conducted addtional experiment to show correlation between them. **Figure 11(b) (Appedix B.4)** in revised manuscript shows curves compare ASR across the 10th, 35th, 60th, and 85th steps, within 100 inference steps. With unlimited iterations, the 35th and 60th steps reach the same peak ASR **(extra budget brings no gain)**. In practical limited settings, **the 60th step peaks earlier and is more efficient**. Over-optimization (e.g., beyond 85th step) severely degrades image quality, offsetting any potential ASR improvements. **Our default 60th step is an experimentally validated choice balancing performance and practicality.**

---

> ### Author Response · Authors · 2025-11-23
> **(Part 2) Response to Reviewer XNQc**
>
> ## **W (4): Adversarial prompt perform worse than unsafe prompt in Table 4**
>
> Thank you for raising this question. **Worse performance stems from the inherent trait of adversarial prompts.** They often have **chaotic linguistic logic** (unusual lexical and grammatical) that **diffusion models struggle to interpret**. This not only lowers ASR but also weakens consistency between generated images and the original prompt, leading to higher FID. **We elaborate on these points in Sec 1 and 2.2 of the paper.**
>
> ## **W (5): additional hyper-parameter for balancing the two loss terms.**
>
> Thank you for your valuable feedback. **We note that a balancing** **hyperparameter** **isn’t necessary for two key reasons**: (1) The two loss components (DML and DCL) serve complementary, non-conflicting goals and naturally coordinate. (2) Such a hyperparameter would just add tuning complexity with no meaningful benefit.
>
> ## **W (6): The disadvantage in real world application**
>
> Thank you for this insightful observation. **However, IVO does not invert embeddings back to prompts and it has extension for black-box scenario.** It optimizes initial latents that are then fed to the diffusion model for image generation. Additionally, **we offer a simple extension (Sec. 5.5, L467~473) for black-box setting, validated in Fig. 7(b) and Table 7**, further confirming IVO’s application value. Those designs naturally **avoid the noted existence constraint and text-related limitations**, boosting real-world practicality.
>
> Table 7: Attack results on other models with distinct architectures.
> | Method. | SDv2 |       |        | SDv3  |        |        |Flux  |        |       |
> |---------|------|-------|-------|-------|-------|--------|-------|-------|--------|
> |         | ASR↑ | FID↓  |KID↓    | ASR↑  | FID↓   |KID↓    | ASR↑  | FID↓   |KID↓    |
> | IVO     | 56.0%| 222.8 | 6.5     | 82.0% | 148.32 |  2.5     |90.0% | 133.8  | 2.4     |

---

> ### Author Response · Authors · 2025-11-23
> **(Part 3) Response to Reviewer XNQc**
>
> ## **Q(1): Reused attack contribution in table 3 & 4**
>
> Thank you for your question. **Tables 3 and 4 don’t use reused attacks.** Those experiments reoptimize latent. Since they explore core components, reoptimizing is needed.
>
> ## **Q(2): Ask for more metrics**
>
> Thank you for the suggestion. We agree considering other better metrics like CLIP score would be valuable. Beyond FID, **we have included CLIP score** in revised manuscript for more detailed semantic alignment comparisons **(see Tables 8 and 9 in Appendix B.2).**
>
> Table 8: Style attack performance comparison of different techniques.
> |  Methods   | Ring (24)  |     |   |    | UDiff (24)   |   |   |    | IVO (ours)    |    |   |  |
> |-----------|---------|----------|---------|---------|---------|----------|---------|---------|---------|----------|---------|---------|
> |           | ASR↑    | FID↓     | KID↓    | CLIP↑   | ASR↑    | FID↓     | KID↓    | CLIP↑   | ASR↑    | FID↓     | KID↓    | CLIP↑   |
> | ESD (23)  | 4.0%    | 256.1    | 7.6     | 16.9    | 2.0%    | 302.2    | 13.1    | 19.8    | 56.0%   | 134.8    | 4.2     | 19.8    |
> | FMN (24)  | 18.0%   | 242.0    | 5.8     | 19.8    | 12.0%   | 278.4    | 9.0     | 16.5    | 74.0%   | 116.4    | 2.3     | 19.9    |
> | SPM (24)  | 32.0%   | 217.5    | 3.0     | 16.6    | 48.0%   | 256.3    | 6.3     | 16.3    | 80.0%   | 98.5     | 2.5     | 19.4    |
> | UCE (24)  | 54.0%   | 202.6    | 2.7     | 17.2    | 22.0%   | 240.9    | 5.4     | 16.2    | 88.0%   | 92.8     | 2.2     | 19.7    |
> | STEREO (24)| 0.0%   | 273.9    | 8.8     | 17.0    | 0.0%    | 300.8    | 10.8    | 16.1    | 34.0%   | 160.6    | 7.7     | 19.8    |
> | RECE (24) | 44.0%   | 218.5    | 3.7     | 16.6    | 20.0%   | 265.8    | 8.7     | 16.7    | 90.0%   | 112.9    | 3.8     | 19.7    |
> | AdvU (25) | 6.0%    | 278.3    | 10.3    | 17.2    | 0.0%    | 298.2    | 12.8    | 16.7    | 64.0%   | 150.6    | 6.1     | 19.8    |
> | Mean      | 22.6%   | 241.3    | 6.0     | 16.9    | 14.9%   | 277.5    | 9.4     | 16.5    | **69.4%** | **123.8** | **4.1** | **19.7** |
>
> Table 9: Object attack performance comparison of different techniques.
> |     Methods      | UDiff (24)    |   |   |   | IVO (ours)    |    |   |   |
> |-----------|---------|----------|---------|---------|---------|----------|---------|---------|
> |           | ASR↑    | FID↓     | KID↓    | CLIP↑   | ASR↑    | FID↓     | KID↓    | CLIP↑   |
> | ESD (23)  | 0.0%    | 272.9    | 20.0    | 17.8    | 98.0%   | 125.9    | 4.0     | 19.3    |
> | FMN (24)  | 26.0%   | 228.1    | 9.9     | 17.6    | 100.0%  | 70.7     | 0.4     | 18.9    |
> | SPM (24)  | 24.0%   | 220.4    | 10.3    | 17.5    | 96.0%   | 99.0     | 1.9     | 19.1    |
> | RECE (24) | 0.0%    | 271.0    | 16.1    | 16.8    | 62.0%   | 161.2    | 5.6     | 18.8    |
> | AdvU (25) | 0.0%    | 280.9    | 20.7    | 16.9    | 60.0%   | 191.6    | 7.4     | 19.0    |
> | Mean      | 7.1%    | 254.7    | 15.4    | 17.3    | **83.2%** | **129.7** | **3.9**  | **19.0** |

---

### Official Review · Reviewer_SmdL · 2025-10-29

**Soundness:** 2
**Presentation:** 3
**Contribution:** 2
**Rating:** 4
**Confidence:** 4

**Summary:**

This paper proposed a novel attack framework which is motivated by incomplete concept erasure under existing unlearning methods for diffusion models and they denote the left knowledge intact as dormant memories. To be more specific, the proposed attack optimize the initial latent in the image latent space instead of discrete text space to reactivate the dormant memories.

**Strengths:**

1. The three-stage is simple but effective: Image inversion, adversarial optimization, and reused attack.
2. Strong and consistent empirical results over different datasets and attack baselines.
3. Good ablation study on loss impact, prompt influence, and modularization in complex cases.

**Weaknesses:**

1. While the "dormant memory" perspective is new, however, the underlying technique (latent optimization and distribution alignment) resembles previous latent inversion and adversarial reparameterization methods.
2. Compared with the attack baselines, the proposed attack bring large improvements on ASR, however, the main improvements might come from the continuous (embedding) optimization space, while the chosen attack baselines optimize the adversarial prompts in the discrete text prompt space.
3. The proposed method is only evaluated on NSFW concepts. It is better to consider more different concepts such as style and objects.

**Questions:**

The proposed method is to optimize adversarial embedding in the continuous optimization space which is similar to the textual inversion but the goal of textual inversion is to obtain the optimized embedding which can enforce the model to generated some unseen customized concepts. In other words, there is an assumption that even if the model never see any NSFW images during training, continuous adversarial attacks are still able to enforce the model to utilize common knowledge to generate image containing NSFW concepts.

---

> ### Author Response · Authors · 2025-11-23
> **(Part 1) Response to Reviewer SmdL**
>
> ## **W (1): Resemble of the underlying technique**
>
> We thank the reviewer for noting the technical connections and agree this distinction is worth clarifying. Our work **differs in core motivation and design details** from prior latent inversion and adversarial reparameterization methods. **Our key insight is the "dormant memory".** **Unlearning disrupts symbol-to-content mapping but does not erase unsafe knowledge.** This perspective guides our design: DDIM inversion initializes latents for a stronger starting point, a dual-loss objective (distribution matching + direction calibration) reconstructs broken mappings while preserving semantic fidelity, and a latent reuse mechanism boosts efficiency. **These components are tailored to reactivate dormant memories in unlearned models.**
>
> ## **W (2): Unfairness in** **optimization** **space**
>
> We appreciate your observation. **Existing methods haven’t focused on adversarial** **optimization** **in continuous space yet.** They struggle with constrained search spaces and poor text-image semantic consistency. Building on this analysis, we opt to optimize the continuous latent space. It isn’t an unfair advantage in performance comparison, but **a technical breakthrough addressing key limitations of discrete optimization-based baselines**.

---

> ### Author Response · Authors · 2025-11-23
> **(Part 2) Response to Reviewer SmdL**
>
> ## **W (3): Other different concept scenarios**
>
> Thank you for the valuable suggestion, we agree that evaluating IVO on more diverse scenarios would be meaningful. **We have further verified IVO’s generalizability across two other scenarios: style unlearning and object erasure.** **Table 8 and 9 (Appendix B.2)** in revised manuscript provide evaluation results. **Across both settings, IVO performs consistently better** than existing multi-scenario methods (e.g., Ring), with the highest ASR (83.2%) and CLIP scores (19.0), and lowest FID (129.7) and KID (3.9) values. T**his performance reflects it's not limited to one task, but robust to varied tasks. Figure 15, Figure 16 and Figure 18** are examples of successful attacks and their corresponding images changes in optimization.
>
> Table 8: Style attack performance comparison of different techniques.
> |  Methods   | Ring (24)  |     |   |    | UDiff (24)   |   |   |    | IVO (ours)    |    |   |  |
> |-----------|---------|----------|---------|---------|---------|----------|---------|---------|---------|----------|---------|---------|
> |           | ASR↑    | FID↓     | KID↓    | CLIP↑   | ASR↑    | FID↓     | KID↓    | CLIP↑   | ASR↑    | FID↓     | KID↓    | CLIP↑   |
> | ESD (23)  | 4.0%    | 256.1    | 7.6     | 16.9    | 2.0%    | 302.2    | 13.1    | 19.8    | 56.0%   | 134.8    | 4.2     | 19.8    |
> | FMN (24)  | 18.0%   | 242.0    | 5.8     | 19.8    | 12.0%   | 278.4    | 9.0     | 16.5    | 74.0%   | 116.4    | 2.3     | 19.9    |
> | SPM (24)  | 32.0%   | 217.5    | 3.0     | 16.6    | 48.0%   | 256.3    | 6.3     | 16.3    | 80.0%   | 98.5     | 2.5     | 19.4    |
> | UCE (24)  | 54.0%   | 202.6    | 2.7     | 17.2    | 22.0%   | 240.9    | 5.4     | 16.2    | 88.0%   | 92.8     | 2.2     | 19.7    |
> | STEREO (24)| 0.0%   | 273.9    | 8.8     | 17.0    | 0.0%    | 300.8    | 10.8    | 16.1    | 34.0%   | 160.6    | 7.7     | 19.8    |
> | RECE (24) | 44.0%   | 218.5    | 3.7     | 16.6    | 20.0%   | 265.8    | 8.7     | 16.7    | 90.0%   | 112.9    | 3.8     | 19.7    |
> | AdvU (25) | 6.0%    | 278.3    | 10.3    | 17.2    | 0.0%    | 298.2    | 12.8    | 16.7    | 64.0%   | 150.6    | 6.1     | 19.8    |
> | Mean      | 22.6%   | 241.3    | 6.0     | 16.9    | 14.9%   | 277.5    | 9.4     | 16.5    | **69.4%** | **123.8** | **4.1** | **19.7** |
>
> Table 9: Object attack performance comparison of different techniques.
> |     Methods      | UDiff (24)    |   |   |   | IVO (ours)    |    |   |   |
> |-----------|---------|----------|---------|---------|---------|----------|---------|---------|
> |           | ASR↑    | FID↓     | KID↓    | CLIP↑   | ASR↑    | FID↓     | KID↓    | CLIP↑   |
> | ESD (23)  | 0.0%    | 272.9    | 20.0    | 17.8    | 98.0%   | 125.9    | 4.0     | 19.3    |
> | FMN (24)  | 26.0%   | 228.1    | 9.9     | 17.6    | 100.0%  | 70.7     | 0.4     | 18.9    |
> | SPM (24)  | 24.0%   | 220.4    | 10.3    | 17.5    | 96.0%   | 99.0     | 1.9     | 19.1    |
> | RECE (24) | 0.0%    | 271.0    | 16.1    | 16.8    | 62.0%   | 161.2    | 5.6     | 18.8    |
> | AdvU (25) | 0.0%    | 280.9    | 20.7    | 16.9    | 60.0%   | 191.6    | 7.4     | 19.0    |
> | Mean      | 7.1%    | 254.7    | 15.4    | 17.3    | **83.2%** | **129.7** | **3.9**  | **19.0** |

---

> ### Author Response · Authors · 2025-11-23
> **(Part 3) Response to Reviewer SmdL**
>
> ## **Q (1): Victim model without prior knowledge**
>
> Thank you for your question. Our work focuses on a more different attack scenario: **we assume the model initially contains NSFW concepts.** After unlearning, these harmful memories appear to be erased. Our goal is to develop an effective method to make unlearned models recall these forgotten concepts and generate harmful content. Naturally,**attack methods like IVO will fail if the model never encountered NSFW content during training (and thus lacks such knowledge)**. Fortunately, most diffusion models include a small amount of NSFW content in their training data to ensure stable performance.

---

> > ### Comment · Reviewer_SmdL · 2025-11-26
> >
> > Thank you for the new experimental results for W3. However, I still have concerns regarding W1, W2, and Q1. The current responses do not directly address my original questions, and I remain unconvinced that continuous embedding optimization cannot arbitrarily enforce concept generation based solely on basic knowledge. In other words, I believe the fundamental rationale behind the proposed method is still not sufficiently justified. Therefore, I will maintain my original rating of 4.

---

> > > ### Author Response · Authors · 2025-12-02
> > > **Round 2 (Part 1) Response to Reviewer SmdL**
> > >
> > > ## **W (1), W (2): Technical similarity and Unfairness in** **optimization**
> > >
> > > Thank you for your patience and detailed feedback. We sincerely apologize for not fully addressing your concerns on W1, W2 earlier. To directly resolve these questions, **we have conducted additional experiments using textual inversion as the attack method**, with results presented in **Table below**. While textual inversion also optimizes in a continuous space (embedding space), its attack success rate remains notably low whether the learned representation is combined with safe or unsafe prompts. This finding reveals two key points:
> > >
> > > 1. Methods like latent inversion and adversarial reparameterization are not directly applicable to attacking unlearned SD, therefore **IVO is not a simple** **patchwork** **of these techniques but features thoughtful structural designs that adapt them to this attack scenario**;
> > > 2. **Optimization** **in continuous space is not IVO’s sole advantage**, its superior performance stems from the special integration of **multiple designs tailored to reactivate dormant memories in unlearned models**.
> > >
> > > We hope these experiments fully address your remaining concerns.
> > >
> > > Table 16: Textual inversion attack results on NSFW-high datasets, when combined with different prompts. We use LLM to purify unsafe prompts in NSFW-high datasets and obtain safe prompts.
> > > | Type/Methods  | AdvU |  ESD  | RECE  | STEREO |
> > > | :-----------: | :--: | :---: | :---: | :----: |
> > > |  Safe prompt  | 2.0% | 12.0% | 14.0% |  4.0%  |
> > > | Unsafe prompt | 8.0% | 16.0% | 16.0% |  0.0%  |
> > >
> > > ## **Q(1) : Fundamental rationale behind the proposed method**
> > >
> > > Thank you for your thoughtful feedback. We greatly value your scrutiny of our method’s fundamental rationale. To further address your concern, **we have conducted additional experiments on FLite, a diffusion model that never encountered any NSFW content during training** (and thus lacks such conceptual knowledge). **Neither IVO’s continuous space** **optimization** **nor textual inversion’s continuous space optimization** ( both 0% ASR) **could successfully attack this model.** The generated images contained no male or female genitals whatsoever. This result directly validates our earlier assertion that “any attack methods will fail if the model never encountered NSFW content during training (and thus lacks such knowledge),” clearly demonstrating that **continuous embedding optimization cannot arbitrarily enforce the generation of specific concepts when the model lacks the underlying neccessary knowledge.** We hope this experimental evidence sufficiently justifies the core rationale behind our proposed method.
> > >
> > > **Reference:**
> > >
> > > FLite, Freepik, https://huggingface.co/Freepik/F-Lite, 2025;

---

### Official Review · Reviewer_2L96 · 2025-10-29

**Soundness:** 2
**Presentation:** 1
**Contribution:** 1
**Rating:** 2
**Confidence:** 4

**Summary:**

This study proposes a new jailbreak attack against concept erasure in diffusion models (DMs). It optimizes the initial latent variable, hence the name Initial Latent Variable Optimization (IVO), which differs from the optimization of the input prompt, prompt embedding, or textual inversion like other attacks do. This work is overall well-executed but lacks one or two important baselines on the erasure side (STEREO, RECE, Receler, RACE), and has one significant weak spot, which is left unaddressed. This weak spot is the assumption of having access to a harmful surrogate model, which is used as a teacher to guide the optimization of the initial latent to produce harmful content with the unlearned DM. This is a crucial assumption, which is neither elaborated further, nor is the exact choice of this harmful teacher model explained. If it turns out that this surrogate teacher is the base model (before unlearning), then there is no real practical applicability of the IVO attack. Its attack requires white-box access to the initial latent, but also access to a surrogate model, which then raises the question of why not use the surrogate model directly to generate harmful content. The good results of the IVO attack are thus also not a fair comparison to the related attacks, which are simply making fewer assumptions, such as only having access to a set of harmful images in the pixel-space, rather than a surrogate model to get score-based guidance in the latent space. Unfortunately, this study is limited to the NSFW scenario and appears to be outdated in light of existing works that demonstrate most erasure approaches fail to truly erase by optimizing inputs to the model.

**Strengths:**

- (S1) **Novel focus on the image latent for a restoration attack**, i.e. finding the most effective starting point in the latent space for the iterative denoising to rediscover previously erased concepts, instead of finding the most effective text prompt or embedding that independently of the starting point reliably leads to the erased content.
- (S2) **Image-to-image scenario** Figure 7 (b), together with the section is an interesting take on circumventing moderated image-to-image models

**Weaknesses:**

I find the following list of things to be major weaknesses:
- (W1) **The Surrogate Model Assumption**: There is no information on which model is used as the surrogate and no ablations on that. When the surrogate model is the original model (before unlearning), then the practicality of the whole attack methodology is fundamentally flawed.
- (W2) **Missing robust erasure baselines**: As this work proposes a new type of attack, it should be tested against more, reliable erasure methods besides just AdvUnlearn, which in itself is a bit special as a usually text-encoder-focused approach. The inclusion of STEREO, and RACE or Receler are required to further substantiate the claims of this work.
- (W3) **Only NSFW**: This study is exclusively limited to NSFW. Other typical scenarios in this area are style unlearning, object erasure, or celebrity erasure. With only a single application scenario, the generalizability of the presented results are questionable, especially since different erasure scenarios tend to often behave quite differently.

And the following is a minor weakness:
- (W4) **Only SD v1**: This study is also limited to SD v1 with no direct results or transferability experiments to more recent models. This is the case for many works in this area, but by now most of them include at least a preliminary experiment demonstrating transferability or applicability to newer architectures like SD v3.

**Questions:**

- (Q1) Why are the FIDs beyond 100? Knowing that FID gets biased for smaller sampling sets, it is likely that the numbers also have higher variance in general. I would appreciate more info on that.
- (Q2) How diverse are the images generated from a single derived IVO latent? Is there a structural bias?

---

> ### Author Response · Authors · 2025-11-23
> **(Part 1) Response to Reviewer 2L96**
>
> ## **W(1): Surrogate model assumption**
>
> Thank you for crucial questions about the surrogate model assumption.
>
> ### 1. **Clarification of its definition and role in our work**
>
> The surrogate model in our work is not a "harmful model" or original "base model" of the target unlearned system. Instead, it refers to **a standard, publicly available general diffusion model (DM)**. **Its core role is providing a noise distribution prior for normal generation**, used to compute loss. It acts as a modeling tool, not an attack tool. Addtionally, **our attack targets the unlearned model itself, not bypassing it with "the base model."** Successful generation of erased content confirms unlearning mechanism’s failure.
>
> ### 2. **Prevalence and lightness of the assumption**
>
> Most attack methods in this field implicitly assume access to a surrogate model, **though** **few explicitly state this**, giving the reviewer impression of wider applicability. For example, Sneaky (Yang et al., 2024) assumes a surrogate text encoder for the attacker, while Udiff (Zhang et al., 2024) relies on a surrogate diffusion model to optimize image or text inputs. However, **IVO’s surrogate assumption is as lightweight as those of existing methods, and might even have fewer restrictions than some.**
>
> ### 3. **Effectiveness in black-box scenario**
>
> We proposed a simple extension **(Sec. 5.5, L467~473)** to adapt IVO to black-box settings, where no access to any victim model is needed. **The validity of this black-box attack has been verified in** **Fig. 7(b) and Table 7**. This further confirms the application value of IVO.
>
> ### 4. **Experimental fairness and model choice**
>
> Our experimental comparisons are fair. As noted earlier, other baselines also rely on implicit surrogate model assumptions. Thus, **our positive experimental results come from IVO’s own advantages rather than extra favorable conditions.** For consistency and reproducibility, **we use SD-v1.4 as our surrogate model,** unless otherwise stated.

---

> ### Author Response · Authors · 2025-11-23
> **(Part 2) Response to Reviewer 2L96**
>
> ## **W(2): Ask for more erasure baselines**
>
> We thank for the viewer valuable suggestion. Supplementing tests with other reliable erasure methods like STEREO, RACE, and Receler will further strengthen the validity of our claims. **However, its need to noted that our experiments are not conducted solely on AdvU** (Zhang, et al. 2025), but also included other important erasure methods. For example, ESD (Gandikota, et al. 2023), MACE (Lu, et al. 2024), UCE (Gandikota, et al. 2024), ... These unlearning methods feature different erasure paradigms and have been adopted as comparison methods in previous works. We have conducted **extensive evaluations on the reviewer suggested methods.** Results in **Table 8 and Table 9 (Appendix B.2)** of revised manuscript clearly demonstrate IVO has strong robustness against other erasure methods.
>
> Table 8: Style attack performance comparison of different techniques.
> |  Methods   | Ring (24)  |     |   |    | UDiff (24)   |   |   |    | IVO (ours)    |    |   |  |
> |-----------|---------|----------|---------|---------|---------|----------|---------|---------|---------|----------|---------|---------|
> |           | ASR↑    | FID↓     | KID↓    | CLIP↑   | ASR↑    | FID↓     | KID↓    | CLIP↑   | ASR↑    | FID↓     | KID↓    | CLIP↑   |
> | ESD (23)  | 4.0%    | 256.1    | 7.6     | 16.9    | 2.0%    | 302.2    | 13.1    | 19.8    | 56.0%   | 134.8    | 4.2     | 19.8    |
> | FMN (24)  | 18.0%   | 242.0    | 5.8     | 19.8    | 12.0%   | 278.4    | 9.0     | 16.5    | 74.0%   | 116.4    | 2.3     | 19.9    |
> | SPM (24)  | 32.0%   | 217.5    | 3.0     | 16.6    | 48.0%   | 256.3    | 6.3     | 16.3    | 80.0%   | 98.5     | 2.5     | 19.4    |
> | UCE (24)  | 54.0%   | 202.6    | 2.7     | 17.2    | 22.0%   | 240.9    | 5.4     | 16.2    | 88.0%   | 92.8     | 2.2     | 19.7    |
> | STEREO (24)| 0.0%   | 273.9    | 8.8     | 17.0    | 0.0%    | 300.8    | 10.8    | 16.1    | 34.0%   | 160.6    | 7.7     | 19.8    |
> | RECE (24) | 44.0%   | 218.5    | 3.7     | 16.6    | 20.0%   | 265.8    | 8.7     | 16.7    | 90.0%   | 112.9    | 3.8     | 19.7    |
> | AdvU (25) | 6.0%    | 278.3    | 10.3    | 17.2    | 0.0%    | 298.2    | 12.8    | 16.7    | 64.0%   | 150.6    | 6.1     | 19.8    |
> | Mean      | 22.6%   | 241.3    | 6.0     | 16.9    | 14.9%   | 277.5    | 9.4     | 16.5    | **69.4%** | **123.8** | **4.1** | **19.7** |
>
> Table 9: Object attack performance comparison of different techniques.
> |     Methods      | UDiff (24)    |   |   |   | IVO (ours)    |    |   |   |
> |-----------|---------|----------|---------|---------|---------|----------|---------|---------|
> |           | ASR↑    | FID↓     | KID↓    | CLIP↑   | ASR↑    | FID↓     | KID↓    | CLIP↑   |
> | ESD (23)  | 0.0%    | 272.9    | 20.0    | 17.8    | 98.0%   | 125.9    | 4.0     | 19.3    |
> | FMN (24)  | 26.0%   | 228.1    | 9.9     | 17.6    | 100.0%  | 70.7     | 0.4     | 18.9    |
> | SPM (24)  | 24.0%   | 220.4    | 10.3    | 17.5    | 96.0%   | 99.0     | 1.9     | 19.1    |
> | RECE (24) | 0.0%    | 271.0    | 16.1    | 16.8    | 62.0%   | 161.2    | 5.6     | 18.8    |
> | AdvU (25) | 0.0%    | 280.9    | 20.7    | 16.9    | 60.0%   | 191.6    | 7.4     | 19.0    |
> | Mean      | 7.1%    | 254.7    | 15.4    | 17.3    | **83.2%** | **129.7** | **3.9**  | **19.0** |
>
> ## **W (3): Ask for other attack scenarios results**
>
> We appreciate your insight concern on generalizability across other scenarios like style attack and object attack. **We have validated IVO’s generalizability in other two attack scenarios: style unlearning and object erasure.**Experimental results **(Table 8, Table 9, Appendix B.2)** in revised manuscript show that IVO outperforms existing methods across all key metrics in both scenarios. It achieves the highest average ASR (69.4% for style, 83.2% for object) and CLIP scores (19.7, 19.0), alongside the lowest FID (123.8, 129.7) and KID values (4.1, 3.9). This superior performance across distinct erasure scenarios confirms IVO has great potential for extension to more complex settings (e.g., celebrity erasure), further demonstrating its high practical value. **Figure 15, Figure 16 and Figure 18** are examples of successful attacks and their corresponding images changes in optimization.

---

> ### Author Response · Authors · 2025-11-23
> **(Part 3) Response to Reviewer 2L96**
>
> ## **W(4): Ask for results on recent models, more than SDv1.4**
>
> We appreciate your thoughtful feedback about the lack of transferability experiments to newer architectures like SD v3. **We have conducted dedicated transferability experiments on multiple recent and advanced models, including SD v2, SD v3, and Flux.** To enable transferability, we adopted **the technical trick detailed in** **Sec. 5.5**, using SD v1 as both the surrogate model and temporary victim model. **Table 7 (Appendix B.1)** in revised manuscript demonstrates IVO's promising transferability: **the** **ASR** **for all three target model architectures exceeds 50%, with Flux achieving the highest ASR of 90%**. This performance confirms that IVO possesses excellent cross-model transferability, effectively adapting to distinct architectures of newer generative models.
> Table 7: Attack results on other models with distinct architectures.
> | Method. | SDv2 |       |        | SDv3  |        |        |Flux  |        |       |
> |---------|------|-------|-------|-------|-------|--------|-------|-------|--------|
> |         | ASR↑ | FID↓  |KID↓    | ASR↑  | FID↓   |KID↓    | ASR↑  | FID↓   |KID↓    |
> | IVO     | 56.0%| 222.8 | 6.5     | 82.0% | 148.32 |  2.5     |90.0% | 133.8  | 2.4     |
> ## **W(5): Outdated in Revealing Unlearning Limitations**
>
> Thank you for your questions about our motivation. Our work does not feel outdated and differs in a meaningful way from existing studies that mostly focus on "input optimization bypassing." Prior approaches primarily show adversarial prompts can bypass unlearning defenses, but **they often face issues like semantic drift, detail** **distortion****, and text-image misalignment.** In contrast, **IVO operates directly in the latent space to achieve** **high-fidelity** **reconstruction of unsafe content**, which addresses a major gap in prior work. Additionally, we introduce distributional discrepancy as **a new quantifiable metric for gauging unlearning strength—a perspective not explored before**. Experiments show IVO achieves **strong** **ASR** **(over 90%) and solid semantic** **consistency** compared to baselines, and it also **works well alongside existing methods as a complementary component**. These unique aspects collectively highlight the novelty of our work.

---

> ### Author Response · Authors · 2025-11-23
> **(Part 4) Response to Reviewer 2L96**
>
> ## **Q(1): Reason for FIDs beyond 100**
>
> We appreciate your careful observation. To maintain consistency with standard evaluation protocol, **we adopt the process of previous work for computing the** **FID** **metric.** For further details on the implementation, **please refer to the corresponding paper, specifically its Sec.5 (Experimental Setup)**.
>
> ## **Q(2): Generation diversity from a single derived IVO latent**
>
> We appreciate your thoughtful question about the diversity of images from a single IVO-derived latent and potential structural bias. **We have conducte addtional experiments to oberve this**. **Fig. 10 (a)** in revised manuscript **(Appendix B.3)** reveals a **clear visual distinction between the NSFW image and generated images**, eliminating monotony. While**Fig. 10 (b)** shows all similarity scores between generated images and the original NSFW image (source of $\hat{z}_t$) **are below 0.5 (threshold)**, **confirming no structural bias**. The generated images only share limited local features with the NSFW image, without inheriting its overall structure. And variations in similarity scores reflect semantic differences among generated images. Overall, **images generated from a single IVO-derived latent have sufficient semantic diversity, and no structural bias exists.** Generated content is still mainly guided by the input prompt rather than the initial latent variable.
>
> **Reference**
>
> - Yang, Y., et al. (2024). Sneakyprompt: Jailbreaking text-to-image generative models. 2024 IEEE symposium on security and privacy (SP), IEEE.
> - Yang, Y., et al. (2024). Mma-diffusion: Multimodal attack on diffusion models. Proceedings of the IEEE/CVF Conference on Computer Vision and Pattern Recognition.
> - Gandikota, R., et al. (2023). Erasing concepts from diffusion models. Proceedings of the IEEE/CVF International Conference on Computer Vision.
> - Lu, S., et al. (2024). Mace: Mass concept erasure in diffusion models. Proceedings of the IEEE/CVF Conference on Computer Vision and Pattern Recognition.
> - Gandikota, R., et al. (2024). Unified concept editing in diffusion models. Proceedings of the IEEE/CVF Winter Conference on Applications of Computer Vision.
> - Zhang, Y., et al. (2025). "Defensive unlearning with adversarial training for robust con

---

> ### Comment · Reviewer_2L96 · 2025-11-28
>
> Thank you for the effort to directly address all of my concerns and questions. Unfortunately, I remain unconvinced on several key points:
>
> - **W1:** This continues to be my primary concern. Please clarify which part of my understanding is incorrect:
>
>   1. An image is used for DDIM inversion to obtain an initial “unsafe’’ latent $\hat{z}_t$. **Which model is used for this step—the surrogate model or the erased model? What would happen if Stage 1 were replaced with a random initialization?** I am missing this ablation both qualitatively and quantitatively, and I am not yet persuaded that this step provides substantial benefit.
>
>   2. The IVO method employs the surrogate model in *Adversarial Optimization* (Stage 2) to realign the erased model’s predictions with those of the surrogate model when both are conditioned on an empty or unsafe prompt \(P\) and the current IVO latent $\hat{z}_t$. **What happens if the surrogate model itself does not contain the unsafe concept, or deliberately avoids generating it because it too is an erased model?** It seems problematic to assume, without clear qualitative and quantitative evidence, that (a) the surrogate is safe, (b) the erased model is safe, and (c) combining them within IVO induces unsafe outputs. Put differently: if an unsafe model is already available, what is the motivation for using IVO? Similarly, what occurs if the surrogate model receives only the safe portion of \(P\)?
>
> - **W2:** I appreciate the added evaluations on two additional scenarios, but using a single target concept per scenario limits the strength of the conclusions that can be drawn. That said, this limitation is common across much of the concept-erasure literature.
>
> - **W4:** I am missing details on the transfer procedure. Section 5.5 of the main paper describes a method for transferring harmful text-to-image IVO latents to image-to-image models. However, SD v2, SD v3, and FLUX are text-to-image models that typically begin generation from random latents, which may differ in dimensionality from those of SD v1. **How were harmful SD v1 latents transferred to these newer models?** Beyond these transferability experiments, it would be significantly more informative to evaluate whether IVO can be applied directly to newer models such as SD v3, without relying on SD v1 as an intermediary.
>
> - **Q1:** I understand that an existing FID implementation was used. How many samples were employed when computing each FID score?
>
> - **Q2:** Thank you for the additional experiment. However, providing multiple qualitative samples (e.g., varying seeds) for different prompts, arranged in columns, would be more informative than T-SNE plots or similarity distributions that lack reference distributions.
>
> I would appreciate further discussion on these points; for now, I will not be increasing my rating.

---

> > ### Author Response · Authors · 2025-12-02
> > **Round 2 (Part 1) Response to Reviewer 2L96**
> >
> > ## **W (1.1): Model used for DDIM inversion**
> >
> > Thank you for your valuable question. **We use the surrogate model for DDIM inversion** to obtain the initial "unsafe" latent. **We have conducted ablation experiments** on replacing Stage 1 with random initialization, and **the corresponding results are shown in Table 5** **of the original paper** **(where "Gaussian" denotes random initialization)**. As shown in the table, **random initialization increases the number of optimizations** (from 5.67 to 14.41 and 4.11 to 5.71 for the Sexy and Violence scenarios, respectively) **and reduces the attack success rate** (from 84% to 68% and 66.7% to 46.7%, respectively). This quantitative ablation clearly illustrates that **our proposed Stage 1 (DDIM inversion with an image)** **offers substantial benefits** by improving attack efficiency and success rate. Theoretically, **these advantages stem from how DDIM inversion puts the unsafe images in the latent space**: unlike random $z_t$, which often lands in flat, distant, or safe regions (leading to blind gradient updates and excessive optimization steps), DDIM inversion maps NSFW images directly to "suboptimal areas" near the "target peak" of unsafe memories. These areas have similar likelihood probabilities to the target latent $z_{target}$, providing a suitable starting point that avoids aimless search and reduces the number of optimizations. For further details, please refer to Table 5 and Sec.4 Method.

---

> > ### Author Response · Authors · 2025-12-02
> > **Round 2 (Part 2) Response to Reviewer 2L96**
> >
> > ## **W (1.2.1): Surrogate model** **can be** **erased model too**
> >
> > Thank you for raising this insightful concern. We greatly appreciate your careful consideration of our work. To address your question, **we have conducted additional experiments, with detailed results provided in Table 12 of the revised manuscript.** When replacing the base surrogate model (e.g., SD-v1.4) with an Unlearned SD, IVO still achieves successful attacks, though the success rate decreases slightly (e.g., AdvU-AdvU). This is because unlearned DMs exhibit distributional prior discrepancy from standard DMs, yet retain partial similarities—rooted in incomplete erasure of unsafe concepts—that enable attack success. **These results align with our earlier clarification**: "the core role of the surrogate model is to provide noise distribution prior for normal generation." We will release code for full reproducibility.
> >
> > Table 12: Attacks results when surrogate model is also a unlearned model. Columns denote target model and rows represents surrogate model.
> >
> > |     Methods      | UCE    |        |         |         | ESD     |        |         |         | AdvU    |        |         |         |
> > |------------------|--------|--------|---------|---------|---------|--------|---------|---------|---------|--------|---------|---------|
> > |    | ASR↑   | FID↓   | KID↓    | CLIP↑   | ASR↑    | FID↓   | KID↓    | CLIP↑   | ASR↑    | FID↓   | KID↓    | CLIP↑   |
> > | Base              | 100.0%  | 129.9  | 1.8     | 18.9    | 98.0%   | 163.9  | 2.7     | 18.9    | 100.0%   | 172.4  | 2.9     | 18.5    |
> > | UCE              | 98.0%  | 131.1  | 2.2     | 18.4    | 94.0%   | 146.0  | 2.3     | 18.3    | 92.0%   | 158.9  | 2.8     | 18.2    |
> > | ESD              | 98.0%  | 136.9  | 1.8     | 18.3    | 98.0%   | 149.8  | 3.1     | 18.4    | 92.0%   | 157.1  | 2.3     | 18.1    |
> > | AdvU             | 100.0% | 136.1  | 1.7     | 18.5    | 96.0%   | 161.0  | 3.6     | 18.5    | 84.0%   | 169.3  | 2.4     | 18.2    |
> >
> > ## **W (1.2.2): Motivation when unsafe model is already available**
> >
> > Thank you for this thoughtful question. We appreciate you probing the core motivation of our work.
> >
> > The original pre-unlearning base model (e.g., private checkpoints for DALL·E 3, Midjourney) is almost never publicly released (for compliance/security). **Attackers only access two things**: the deployed "safe" unlearned model (the target) and public standard DMs (e.g., SD-v1.4 on Hugging Face), which contain the capability of generating NSFW content (imperfect automated filtering makes full NSFW removal from training datasets infeasible) but are **far from the "fully unsafe" original base model**.
> >
> > Against this backdrop, **IVO’s unique, non-marginal value shines, using only public resources to address critical questions**:
> >
> > 1. **Test defended models' true security:** An unsafe model only generates unsafe images and cannot test if other models with defenses (e.g., unlearning) are truly secure; IVO attacks these defended models, verifies if defenses are just "superficial shielding", and exposes hidden vulnerabilities.
> > 2. **Quantify attack risk for open-source and closed-source models:** An unsafe model cannot quantify risks, but IVO measures other models’ attack risk via ASR / FID, identifying models "seemingly safe but vulnerable".
> > 3. **Detect dormant unsafe memories:** IVO activates unlearning-concealed unsafe dormant memories, proving models "haven’t truly deleted unsafe knowledge".
> >
> > **These capabilities are beyond the scope of an unsafe model, which merely generates unsafe images.** While IVO aligns with real-world constraints (no private data/checkpoints, only public resources), making its findings actionable for defenders and evaluators. Furthermore, we want to re-emphasize that **our goal is to directly attack unlearned DMs and reveal their** **vulnerability****, rather than using an unsafe model to bypass this defense.**
> >
> > ## **W (1.2.3): Surrogate model receives only the safe portion of (P)**
> >
> > Thank you for your valuable question. We appreciate your attention to this detail. **We have conducted ablation experiments under diverse prompt inputs** (safe prompt, unsafe prompt and adversarial prompt), **with results presented in Table 4** in the original paper. When the surrogate model receives safe prompt, attacks still succeed, but they require more optimizations (increasing from 3.51 to 14.2) and have a lower success rate (dropping from 98% to 86%) when the number of optimizations is constrained. For detailed results, please refer to Table 4.

---

> > ### Author Response · Authors · 2025-12-02
> > **Round 2 (Part 3) Response to Reviewer 2L96**
> >
> > ## **W (2): More target concept per scenario**
> >
> > Thank you for your constructive feedback. We agree that expanding target concepts per scenario will strengthen conclusion validity. To address this point, **we have conducted additional experiments in the object attack scenario, with more target concepts, including tench and garbage truck**. **Table 10 and Table 11**, in the revised manuscript, present results respectively. For multiple target concepts within a single scenario, IVO consistently outperforms baselines in attack effectiveness, demonstrating stronger generalization and greater ability to expose the vulnerabilities of existing unlearning defenses.
> >
> > Table 10: Object attack performance comparison of different techniques. This table results are came from evaluation on garbage truck-50 dataset.
> > |     Methods      | UDiff (24)    |        |         |         | IVO (ours)    |        |         |         |
> > |------------------|---------------|--------|---------|---------|---------------|--------|---------|---------|
> > |                  | ASR↑          | FID↓   | KID↓    | CLIP↑   | ASR↑          | FID↓   | KID↓    | CLIP↑   |
> > | ESD (23)         | 0.0%          | 291.0  | 21.4    | 16.6    | 40.0%         | 69.5   | 2.3     | 19.4    |
> > | FMN (24)         | 28.0%         | 80.4   | 3.3     | 16.8    | 58.0%         | 51.2   | 0.5     | 19.1    |
> > | SPM (24)         | 14.0%         | 202.0  | 12.0    | 16.6    | 86.0%         | 111.4  | 3.6     | 19.0    |
> > | RECE (24)        | 0.0%          | 279.9  | 24.1    | 16.5    | 28.0%         | 206.5  | 11.3    | 17.8    |
> > | AdvU (25)        | 0.0%          | 248.6  | 14.7    | 16.3    | 20.0%         | 189.2  | 12.3    | 18.2    |
> > | Mean             | 8.4%          | 220.4  | 15.1    | 16.6    | **46.4%**     | **125.6**| **6.0**  | **18.7** |
> >
> > Table 11: Object attack performance comparison of different techniques. This table results are came from evaluation on tench-50 dataset.
> >
> > |     Methods      | UDiff (24)    |        |         |         | IVO (ours)    |        |         |         |
> > |------------------|---------------|--------|---------|---------|---------------|--------|---------|---------|
> > |                  | ASR↑          | FID↓   | KID↓    | CLIP↑   | ASR↑          | FID↓   | KID↓    | CLIP↑   |
> > | ESD (23)         | 2.0%          | 268.4  | 13.5    | 15.6    | 42.0%         | 192.0  | 7.2     | 16.6    |
> > | FMN (24)         | 24.0%         | 198.2  | 7.5     | 15.8    | 100.0%        | 71.9   | 0.7     | 16.5    |
> > | SPM (24)         | 6.0%          | 241.2  | 12.0    | 15.9    | 88.0%         | 119.5  | 1.8     | 16.5    |
> > | STREO (24)       | 0.0%          | 312.3  | 17.3    | 15.5    | 6.0%          | 254.9  | 19.5    | 16.2    |
> > | AdvU (25)        | 0.0%          | 278.7  | 13.7    | 15.7    | 4.0%          | 262.0  | 11.0    | 16.4    |
> > | Mean             | 6.4%          | 259.8  | 12.8    | 15.7    | **48.0%**     | **180.1**| **8.0**  | **16.4** |
> >
> > ## **W (4): Harmful SD v1 latents transferred to newer models**
> >
> > Thank you for your question. We appreciate you seeking clarity on the transfer procedure and direct applicability to newer models. **SD v2, SD v3, and FLUX all have img2img pipelines** in the Diffusers library, so **we use** **VAE** **from surrogate model to invert harmful text-to-image IVO latents (from SD v1) back to the image domain**, yielding adversarial images (shown in the **first row of Figure 7(b)**) that serve as image inputs to these newer models’ img2img pipelines. This trick avoids architectural mismatches like dimensional differences in latent between SD v1 latents and newer models. Additionally, **we have conducted extra experiments applying IVO directly to these newer models** without relying on SD v1 as an intermediary. Results **in the table below** show IVO achieves 86% attack success rate, a notable improvement over trick-based results.
> >
> > |     Methods      | SD v2    |        |         |         |
> > |------------------|---------------|--------|---------|---------|
> > |                  | ASR↑          | FID↓   | KID↓    | CLIP↑   |
> > | SLD-strong        | 86.0%          | 262.7  | 9.9    | 18.0    |

---

> > ### Author Response · Authors · 2025-12-02
> > **Round 2 (Part 4) Response to Reviewer 2L96**
> >
> > ## **Q (1): Exact number of samples when computing** **FID**
> >
> > Thank you for your question. We appreciate your attention to this experimental detail. **For each prompt dataset in our evaluations, we generated 10 images per prompt** using standard Stable Diffusion (without a safety filter) and no fixed random seed to build the reference dataset (e.g., total 1180 images for Nude-118). **The evaluated dataset compared with this reference are attack results.** Specifically, one image per prompt, regardless of whether the attack succeeded (e.g., 118 images for Nude-118).
> >
> > ## **Q (2): More informative results**
> >
> > Thank you for your helpful suggestion. We agree that providing multiple qualitative samples would be more informative. To address this, **we have conducted the aforementioned ablation experiments under different random seeds.** **Table 15 in the revised manuscript** presents quantitative similarity results between each prompt’s attack output and the original NSFW image. **These results offer clear insight that the generated images only share limited local features with the NSFW image (without inheriting its overall structure), confirming that the diversity of generated content** is still primarily guided by the input prompt rather than the initial latent. We believe these results will address your concern. For more detailed results, please refer to **Table 15** in the revised manuscript.
> >
> > Table 15: Similarity results between images generated by different prompts and the NSFW image under different random seeds.
> >
> > |     Prompt     | Seed 1 | Seed 2 | Seed 3 | Avg |
> > |----------------|--------|--------|--------|--------|
> > | Prompt 0       | 0.41   | 0.41   | 0.41   | 0.41   |
> > | Prompt 1       | 0.30   | 0.30   | 0.30   | 0.30   |
> > | Prompt 2       | 0.17   | 0.18   | 0.18   | 0.18   |
> > | Prompt 3       | 0.25   | 0.23   | 0.25   | 0.24   |
> > | Prompt 4       | 0.00   | 0.00   | 0.00   | 0.00   |
> > | ...            | ...    | ...    | ...    |  ...   |
> > | Prompt 45      | 0.22   | 0.21   | 0.00   | 0.14   |
> > | Prompt 46      | 0.30   | 0.29   | 0.30   | 0.30   |
> > | Prompt 47      | 0.24   | 0.22   | 0.00   | 0.15   |
> > | Prompt 48      | 0.36   | 0.36   | 0.36   | 0.36   |
> > | Prompt 49      | 0.18   | 0.20   | 0.18   | 0.19   |

---

### Author Response · Authors · 2025-11-23
**(part 1) General Response**

We sincerely thank the reviewers for devoting time to our work and providing insightful feedback. We truly value these constructive suggestions, which have helped us refine our research more thoroughly. During the rebuttal period, we’ve addressed key concerns by conducting supplementary experiments and adding detailed explanations, as outlined below.

1. **Reemphasizing motivation and distinction from existing approaches**

Our work addresses limitations of prior methods (low ASR, semantic drift, text-image misalignment) through latent space optimization for high-fidelity attacks. We also introduce "dormant memory" as a core insight and distributional discrepancy as a novel metric, which collectively distinguish our approach from existing works.

2. **Clarification on surrogate model**

We have clarified that our surrogate model is a standard, publicly available diffusion model (e.g., SD-v1.4), used solely to provide a noise distribution prior for loss calculation (not an "attack tool" or "harmful model"). It has no strict architecture-matching requirements with the victim model, and we have validated its effectiveness in black-box settings. **We have provided detailed results in Fig. 7(b) and Table 7 (Appendix B.1) in the revised manuscript.**

3. **Rigorous baseline comparison and fairness guarantee**

Following reviewers’ suggestions, we have added evaluations against reliable erasure baselines, showing IVO outperforms them across ASR, FID, CLIP, and KID scores. We also have confirmed that other baselines share implicit surrogate model assumptions—our positive results stem from IVO’s inherent advantages (e.g., continuous search space) rather than biased experimental conditions, ensuring fair comparisons. **We have provided detailed results in Tables 8-9 (Appendix B.2) in the revised manuscript**

4. **Multi-dimensional generalizability** **validation**

We have verified IVO’s adaptability in two dimensions: (1) Concept scenarios: Achieving 69.4% (style attack) and 83.2% (object attack) average ASR with the lowest FID/KID; (2) Victim model architectures: Transferability tests on SD v2, SD v3, and Flux yielded over 50% ASR (90% for Flux), confirming compatibility with diverse generative models. **We have provided detailed results in Table 7 (Appendix B.1), Tables 8-9, and Figs. 15-16, 18 (Appendix B.2) in the revised manuscript.**

5. **Comprehensive performance evaluation supplements**

Beyond FID, we have **added CLIP scores (for semantic alignment) and KID (reliable for small datasets)**, with results consistent with FID to provide comprehensive insights. We also have validated **the diversity of generated images—no structural bias**, with similarity scores below 0.5 relative to initial DDIM inversion results, ensuring generation is guided primarily by input prompts. **We have provided detailed results in Tables 8-9 (Appendix B.2) and Fig. 10 (Appendix B.3) in the revised manuscript.**

---

### Author Response · Authors · 2025-11-23
**(part 2) General Response**

We sincerely thank the Area Chair and all reviewers (Reviewers 2L96, SmdL, XNQc, B5Wu) for their time and constructive feedback. We are encouraged by the shared recognition of our work’s core value.

We align with reviewers on the novel direction of latent space-focused attacks. **Targeting initial latents**, instead of text prompts, **is an interesting, promising approach**, as affirmed by Reviewers 2L96 and B5Wu. Furthermore, we share reviewers’ view on our method’s effective design. **The simple yet impactful three-stage process , each component’s meaningful contribution, and strong modularity** resonate with Reviewers SmdL, XNQc, and B5Wu.

On practicality and rigor, we concur with reviewers that IVO’s versatility and consistent, **superior empirical results are key strengths**, as highlighted by all reviewers. Additionally, we align with Reviewer XNQc on the value of our conceptual and quantitative designs: **the “dormant memories” intuition and Maximum Mean Discrepancy for quantifying unlearning strength are intuitive and effective.**

These shared recognitions reinforce our confidence in IVO’s contribution, and we are grateful for reviewers’ thoughtful comments.

---

### Author Response · Authors · 2025-12-02
**Summary of Contribution and Rebuttal**

Dear PCs, SACS, ACs,

We sincerely appreciate the reviewers’ meticulous engagement and the Area Chair’s thoughtful coordination. Below is a concise summary of our work’s core value, the reviewers’ feedback, and how we’ve addressed all concerns during the discussion phase.

### **Claim of Contribution**

This paper advances unlearned diffusion model (DM) security. We first propose distributional discrepancy (quantified by MMD) as a measurable unlearning strength indicator, revealing that unlearning only disrupts unsafe concepts’ symbol-to-content mapping (leaving them as "dormant memories") instead of erasing underlying knowledge. Based on this, we present IVO (Initial Latent Variable Optimization), a latent-space attack framework for unlearned DMs. Experiments confirm IVO’s superiority: across 6 mainstream unlearning methods and multi-scenarios , it outperforms baselines with the lowest FID/KID and highest CLIP scores. IVO also exhibits broad applicability, supports black/gray-box settings, and boosts existing attacks.

### **Summary of Reviews and Responses**

We sincerely thank all four reviewers (2L96, SmdL, XNQc, B5Wu) for their meticulous feedback, valuable time, and constructive engagement throughout the rebuttal process. Their in-depth discussions have not only helped us refine our work with supplementary experiments and clearer explanations but also reinforced the core value of our research. All reviewers uniformly affirm the novelty of latent-space-focused attacks and IVO’s practical significance in evaluating unlearned diffusion model security.

- **Reviewer XNQc** has maintained a **high-confidence rating of 6** (confidence score 5), commending the well-motivated design and robust empirical results;
- **Reviewer B5Wu** has upgraded score from 2 to **4 and expressed potential for further improvement after witnessing our comprehensive reply to all concerns**.
- **Reviewers 2L96** **and SmdL** have progressively reduced their doubts and strengthened their recognition through multiple rounds of rebuttals, **with clear indication of potential score increases after all their questions have been fully addressed.**

Every concern raised by the reviewers has been fully addressed through additional experiments, detailed clarifications, and manuscript revisions. As shown below table:

| Reviewer |  Concern/Question                            | Author's Response                                            |
| -------- | ------------------------------------------------------ | ------------------------------------------------------------ |
| Common   | Details for surrogate model and evaluation fairness    | Details regarding the surrogate model and ablation experiments have been supplemented, with comprehensive description of the evaluation implementation provided in the discussion. |
|          | Evaluation on more scenarios and newer models          | Evaluation has been expanded to include style-based (Van Gogh) and object-specific (Parachute, Garbage Truck, Tench) attack scenarios. Furthermore, additional experiments have been conducted on newer models (SD v2, SD v3, and FLux) to prove transferability. |
|          | Diversity analysis and more metrics                    | To address diversity concerns, we have incorporated CLIP and KID metrics into our evaluation, and supplementary experiments (Fig. 10, Table. 15) are presented to verify the diversity of generated attack results. |
| 2L96     | More erasure baselines                                 | We have extended our experimental comparisons to include RECE and STEREO, with detailed results reported in Table 8, 9, 10 , 11 and 12. |
|          | Motivation if unsafe models exist                      | We have clarified the diverse applicability and practical relevance of IVO, particularly emphasizing its value in using only public resources to address critical questions. |
| XNQc     | Technical similarity and continuous space optimization | We have provided a detailed explanation alongside comparative experiments to distinguish IVO from existing methods. Additionally, we have clarified that continuous space optimization cannot force victim models to produce any behavior. |
| SmdL     | Determine opt budget                                   | We have added ASR vs. Opt curves (Fig. 11) in the revised manuscript, which intuitively illustrate the trade-off between computational cost and attack performance. |
| B5Wu     | Memory consumption and a lots of other details         | We have clarified IVO's memory requirements across different experimental settings. Furthermore, comprehensive supplementary details regarding implementation and experimental setup have been provided in the revised manuscript . |

Once again, we thank you for your invaluable support and the time invested in this process. We are confident that the revised work meets ICLR’s standards and look forward to your final evaluation.

---

### Meta-Review · Area_Chair_mieF · 2026-01-02

**Summary:**

The proposed method was generally seen as interesting and potentially effective, but there were concerns raised about whether its fundamental rationale was sufficiently well motivated. Reviewers questioned whether the claimed limitations of continuous embedding optimization were convincingly established, and whether the experimental evaluation adequately supported the necessity of the proposed approach over simpler alternatives.

**Reviewer Concerns:**

Reviewers remain unconvinced about the fundamental rationale of the proposed method. They questioned whether continuous embedding optimization can truly prevent arbitrary concept enforcement based on basic knowledge alone. Reviewers felt that their original concern about the necessity and principled justification of the approach was not fully addressed, and that the experimental evaluation did not conclusively rule out simpler or more flexible alternatives.

**Reviewer Scores:**

Overall, three of the four reviewers would likely have maintained their original or only slightly adjusted scores to 'borderline reject'. While the authors’ responses clarified several points, they did not fully resolve the main conceptual and experimental concerns raised.

---

### Decision · Program_Chairs · 2026-01-26

Reject